# PTEN, MMP2, and NF-κB and Regulating MicroRNA-181 Aggravate Insulin Resistance and Progression of Diabetic Nephropathy: A Case-Control Study

Manoj Khokhar [1], Purvi Purohit [1,*], Sojit Tomo [1], Riddhi G. Agarwal [2], Ashita Gadwal [1], Nitin Kumar Bajpai [3], Gopal Krishna Bohra [4] and Ravindra Kumar Shukla [5]

[1] Department of Biochemistry, AIIMS, Jodhpur 342005, India
[2] Department of Biological Sciences, Tata Institute of Fundamental Research, Mumbai 400005, India
[3] Department of Nephrology, AIIMS, Jodhpur 342005, India
[4] Department of General Medicine, AIIMS, Jodhpur 342005, India
[5] Department of Endocrinology and Metabolism, AIIMS, Jodhpur 342005, India
* Correspondence: dr.purvipurohit@gmail.com; Tel.: +91-9928388223

**Abstract:** Diabetic nephropathy (DN) is characterized by an increase in urinary albumin excretion, diabetic glomerular lesions, and a decline in glomerular filtration rate (GFR). We assessed the expression of phosphatase and tensin homolog (PTEN), nuclear factor kappa-β (NF-κB), matrix metalloproteinase-2 (MMP2), and microRNA-181 in healthy controls (HC), individuals with type 2 diabetes mellitus (T2DM) without nephropathy, and those with DN. Our study investigated the association between these genes, insulin resistance (IR), and eGFR to gain insight into their roles in the pathogenesis and progression of DN. Anthropometric measurements and biochemical tests were conducted on HC (N = 36), T2DM (N = 38) patients, and DN (N = 35) patients. We used real-time polymerase chain reaction (RT-PCR) for whole blood gene expression analysis and performed bioinformatics analyses, including protein–protein interaction, gene ontology, and co-expression networks. We compared our expression data with other GEO-Microarray datasets. Our study highlights the role of IR in the progression of nephropathy in T2DM via the PTEN-Akt-mTOR signalling pathway. We also observed a decreasing trend in the expression of MMP2 and PTEN and an increasing trend in the expression of NF-κB and miR-181b-5p with the progression of nephropathy to the severe stage. The dysregulated expression of MMP2, PTEN, NF-κB, and miR-181b-5p in patients with T2DM contributes to the progression of T2DM to DN by aggravating IR, inflammation, accelerating basement membrane thickening, mesangial matrix expansion, and renal fibrosis.

**Keywords:** T2DM; diabetic nephropathy; insulin resistance; PTEN; MMP2; NF-κB; miR-181

## 1. Introduction

Diabetic nephropathy (DN) is characterized by an increase in urine albumin excretion with diabetic glomerular lesions and a loss of glomerular filtration rate (GFR). The increased prevalence of diabetes pushed DN to become one of the most common causes of end-stage renal disease (ESRD) all over the world [1]. Genome–wide transcriptome analysis is being used to explore and identify factors that lead to the progression of DN [2]. Similarly, gene expression profiling using microarray analysis also helped to identify several genes of interest that may have a role in the development of DN [3]. The progression of DN from early to advanced stages was also found to be associated with altered genetic expressions of key genes of immune response, inflammation, ion transport, and cell differentiation [4]. Expression profiling and the assessment of plasma microRNAs yielded multiple microRNAs that can distinguish early DN in diabetes patients and contribute to the pathogenesis of DN [5]. Hence, analysing blood microRNAs in diabetic patients, with and without nephropathy, may give an insight into the possible pathways involved

in the progression of DN. The possible role of phosphatase and tensin homolog (PTEN) in the progression of DN, performed by influencing renal tubulointerstitial fibrosis and epithelial-mesenchymal transition, was already discussed by our group [6–8]. In addition, the targeting of PTEN by microRNA-181-5p (hsa-miR-181b-5p) was evidenced in a similar study [9]. Sequentially, PTEN is known to have an inhibitory effect on the expression of matrix metalloproteinase-2 (MMP2) [10]. Although multiple studies were undertaken in diabetes, to our knowledge, no study has yet explored expression of the aforementioned genes together in PBMC (peripheral blood mononuclear cells) in diabetic patients with and without nephropathy. Therefore, we explored the gene expression of the key inflammatory gene nuclear factor kappa-β (NF-κB) along with genes promoting fibrosis (MMP2 and PTEN) and their regulating miR 181b-5p in association with insulin resistance (IR) and estimated glomerular filtration rate (eGFR) to unravel their mechanistic involvement in the progression and pathogenesis of DN.

## 2. Materials and Methods

### 2.1. Study Design and Participants

This study was conducted following the principles of the Declaration of Helsinki for medical research involving human subjects, and ethical approval was obtained from the Institutional Ethics Committee of AIIMS, Jodhpur ("Certificate Reference number: AIIMS/IEC/2019-20/792"). This study employed a random allocation method to assign participants to one of three distinct groups: individuals diagnosed with type 2 diabetes mellitus (T2DM) without nephropathy, those with T2DM with nephropathy (DN), and a healthy control (HC) group. Cases were recruited from either individuals aged between 30 and 80 years with T2DM and a fasting blood sugar (FBS) level >126 mg/dL or individuals who had been diagnosed with diabetes for at least 2 years and had a current haemoglobin A1c (HbA1c) level >6.5%. DN cases were defined as those with urinary microalbumin levels >30 mg/dL in addition to the aforementioned criteria. DN cases were defined as those with urinary microalbumin levels >30 mg/dL and estimated glomerular filtration rate (eGFR) values in addition to the aforementioned criteria. We excluded patients with type 1 diabetes, gestational diabetes, hypothyroidism, coronary artery disease, chronic obstructive pulmonary disease, malignancies, hepatic disorders, haemochromatosis, and any other diagnosed immune disorders from the study (Figure 1).

### 2.2. Biochemical Laboratory Tests

Anthropometric measurements were taken and biochemical tests were carried out on fasting blood samples collected from HC (N = 36), T2DM patients (N = 38), and DN patients (N = 35). HbA1c was measured via latex agglutination inhibition assay. Chemistry assays for the quantitative determination of FBS, low-density lipoprotein (LDL), high-density lipoprotein (HDL), total cholesterol, urea, and creatinine in serum were performed with a Beckman Coulter AU analyser. Serum insulin was analysed by using the chemiluminescence method with a Diasorin Liasion analyser. Urinary microalbumin was estimated by using the immunoprecipitation method. The eGFR of the study population was calculated using the MDRD formula, listed below:

$$\text{GFR (mL/min/1.73 m}^2) = 186 \times (\text{Creatinine}/88.4) - 1.154 \times (\text{Age}) - 0.203 \times (0.742 \text{ if female}) \times (1.210 \text{ if black}).$$

### 2.3. RNA Isolation, Quantification, Reverse Transcription, and Real-Time PCR Expression

Total RNA was isolated from PBMCs using Trizol (RNA-XPress™ Reagent, HiMedia Laboratories, Mumbai, India) and by following the manufacturer's instructions. Samples with 260/280 and 260/230 ratios of ≥1.8 were considered for further downstream processes. RNA was reverse-transcribed using an miScript® II RT Kit (Catalogue no. 218160, Qiagen, Hilden, Germany). The HiFlex Buffer was used to prepare cDNA for real-time PCR, which involved quantifying mature miRNAs as well as mRNAs. The CFX96 Real-Time System and CFX Manager Software (Bio-Rad, Hercules, CA, USA) were used for the real-time

expression analysis of PTEN, NF-κB, and MMP2 in human whole blood samples, which was done using a Dream TaqTM Green PCR Master Mix Kit (Thermo Scientific<sup>TM</sup> Waltham, MA, USA) and the real-time expression analysis of hsa-miR-181b-5p in human whole blood samples was done using an miScript<sup>®</sup> SYBR<sup>®</sup> Green PCR Kit (Qiagen, Hilden, Germany) and a RT2 qPCR Primer Assay Catalogue No: MS00006699, Lot No: 20181300026 ( Qiagen Hilden, Germany) per the manufacturers' instructions. A 10 μL reaction mixture was prepared to detect the real-time expression of hsa-miR-181b-5p, and the samples were run on a 96-well plate under the following conditions: initial activation step at 95 °C for 15 min, followed by 40 cycles of 15 s at 94 °C, 30 s at 55 °C, and 30 s at 70 °C. The real-time PCR reaction was carried out in a PCR machine/thermal cycler to detect hsa-miR-181b-5p levels in the samples. Similarly, a 10 μL reaction mixture was prepared for the detection of PTEN, NF-κB, and MMP2. The samples were run on a 96-well plate under the following conditions: initial activation step at 95 °C for 3 min, followed by 40 cycles of 30 s at 95 °C, 30 s at 55–59 °C (annealing), and 1 min at 72 °C. The real-time PCR reaction was carried out in a PCR machine/thermal cycler to detect PTEN, NF-κB, and MMP2 levels in the samples. The fold change (FC) levels of amplified PTEN, NF-κB, and MMP2 and those of hsa-miR-181b-5p in comparison to those of GAPDH and of RNU6 (internal control/housekeeping gene), respectively, were evaluated using the double delta/ΔΔCT method for cases and controls.

*2.4. In Silico Protein–Protein Interactions and Network Constructions of PTEN, NF-κB, and MMP2 with Interacting Genes*

All proteins that interact with PTEN, NF-κB, and MMP2 were visualized with the help of the STRING Protein–Protein Interaction (PPI) database (version 11.5) [11]. The PPI network was encoded with PTEN-regulated genes and PTEN, NF-κB, and MMP2. We chose a medium-level confidence score criterion (0.400) for the construction of the PPI network. We observed 13 nodes, 35 edges, an average node degree of 5.38, an avg. local clustering coefficient of 0.806, an expected number of edges of 21, and a PPI enrichment *p*-value of 0.002. Cytoscape-v.3.8.2 was used for the construction of the PPI networks [12].

*2.5. Pathway Analysis and Gene Ontology (GO) Enrichment*

Gene ontology (GO) enrichment is a method used to identify the functional categories of genes and their products that are over-represented in a set of differentially expressed genes. GO categorizes genes into three main categories: cellular component (CC), biological process (BP), and molecular function (MF). Enrichment analysis can help us understand the biological functions and processes that are affected by a specific set of genes. In this study, pathway analysis and GO enrichment were performed using STRING (version 11.5) to identify the biological pathways and networks associated with the genes of interest (PTEN, NF-κB, MMP2, and their interacting proteins). The significant pathway analysis results were ranked by their false discovery rates (FDR). Similarly, GO enrichment was performed for CC, BP, and MF related to the genes of interest. These analyses can provide insight into the underlying biological mechanisms and functions of the genes in the context of chronic kidney disease [11].

*2.6. hsa-miR-181b-5p Target Prediction, Network Construction, and Pathways Analysis*

The putative targets of hsa-miR-181b-5p and their interacting genes were predicted using established miRNA target prediction databases such as miRNET, miRanda, Target Scan, and PicTar. hsa-miR-181b-5p was predicted to target the microRNA of PTEN, and its interacting gene, TIMP metallopeptidase inhibitor 3 (TIMP3), by at least three of these databases [13]. A co-expressed network based on PTEN and its interacting gene (TIMP3), the latter of which was targeted by hsa-miR-181b-5p in T2DM and DN, was constructed by using Cytoscape-v3.9.0 [11]. We constructed a network of genes targeted by hsa-miR-181b-5p (PTEN and TIMP3 genes) with other PPI-involved genes. We also analysed the enrichment of microRNA by using MIENTURNET databases [14].

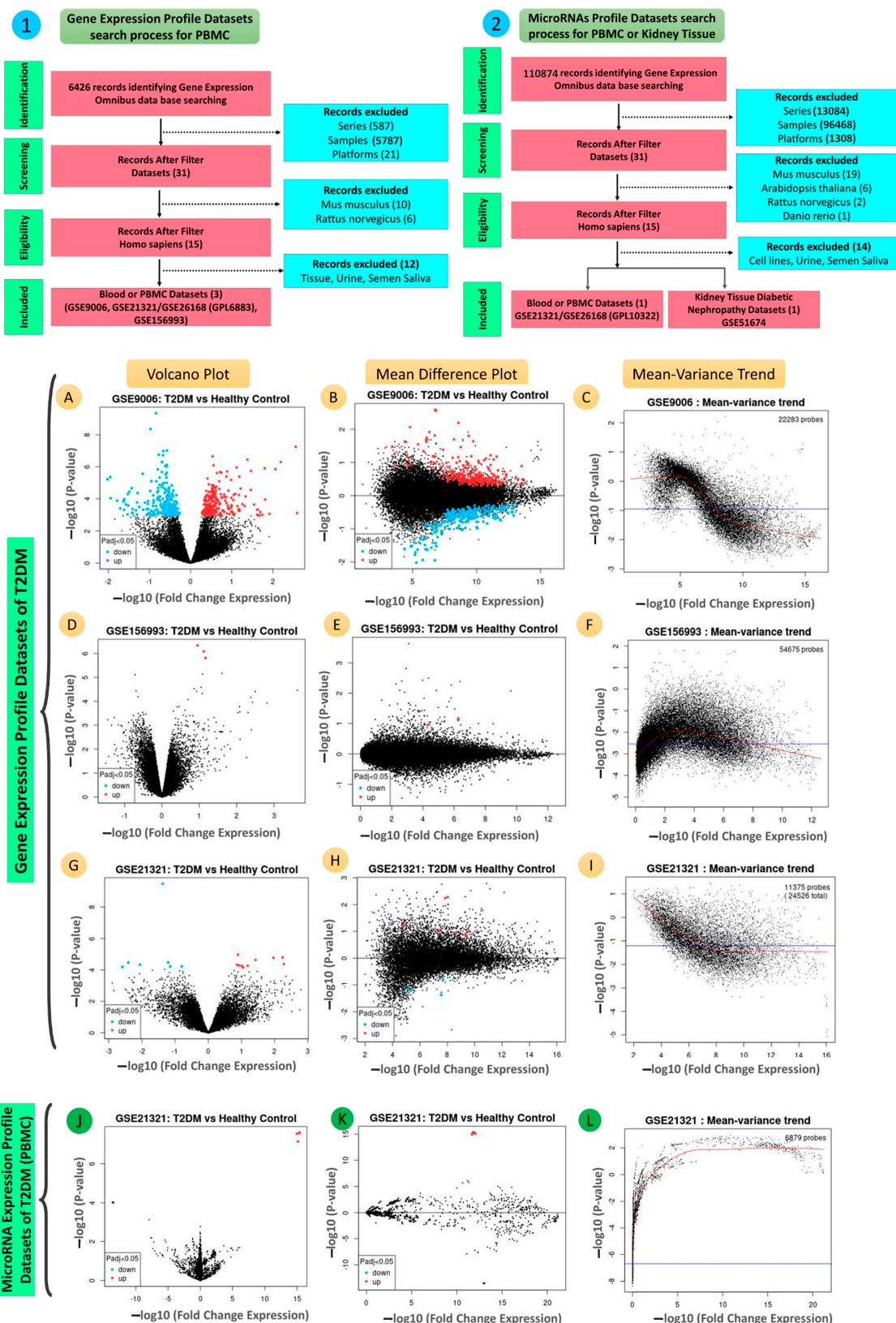

**Figure 1.** The flow diagram illustrates the process of (**1**) the gene expression profile datasets and (**2**) the microRNAs expression profile datasets collections and shows the number of datasets considered for inclusion. (**A,D,G,J**) volcano plots; (**B,E,H,K**) mean difference plots; (**C,F,I,L**) mean–variance trends showing the DEGs in all four datasets, GSE9006, GSE156993, GSE21321/GSE26168 (GPL6883), and GSE21321/GSE26168 (GPL10322), for blood of T2DM.

### 2.7. Gene Expression Profile Dataset Search Process and Inclusion and Exclusion Criteria

We searched several keywords, including "Type 2 Diabetes mellitus", "T2DM", "PBMC", "Blood", "T2DM", "Expression profiling by array", and "Homo sapiens", in the GEO datasets, from which three datasets were selected for this study: GSE9006 [15], GSE156993, and GSE21321/GSE26168 (GPL6883) [16].

The criteria for the selection and collection process and the quality control plots are summarized in Figure 1A–I and Tables S1 and S2. The selected datasets were used for the validation of key genes and other interacting genes in the PPI network.

### 2.8. MicroRNAs Expression Profile Dataset Search Process and Inclusion and Exclusion Criteria

We searched several keywords, including "Type 2 Diabetes mellitus", "T2DM", "PBMC", "Blood", "Tissue", "Renal", "Kidney", "T2DM", "Expression profiling by array", "Homo sapiens", "MicroRNA", "miRNA", and "miR", in the GEO datasets, from which two datasets were selected for this study: GSE21321/GSE26168 (GPL10322) [16] and GSE51674 [17].

The criteria for the selection and collection process and quality control plots are summarized in Figure 1J–L and Tables S3 and S4.

### 2.9. Statistical Analysis

We used IBM SPSS Statistics 23.0 for Windows (SPSS Inc, Chicago, IL, USA) and Orange Data Mining software for performing independent *t*-tests, Bonferroni's post hoc tests, ANOVA tests, and logistic regression analyses. Box and whisker plots were used to evaluate the relationships between variables. Microsoft Excel was used for the graphical representation of the results. *p*-values equal to or less than 0.05 are statistically significant (** $\leq$ 0.05; * $\leq$ 0.001).

## 3. Results

### 3.1. Anthropometric Characteristics of the Study Population

We divided the study population into three groups: healthy controls (HC) (N = 36), type 2 diabetes mellitus (T2DM) patients without nephropathy (N = 38), and T2DM patients with diabetic nephropathy (DN) (N = 35). On assessment, a significant difference ($p < 0.001$) in the WHR (waist-to-hip ratio) was observed between the HC, T2DM, and DN patients. Although DN and T2DM patients were found to be overweight, no statistically significant differences were observed. Furthermore, T2DM and DN patients also had a significant difference in age compared to controls (Table S5).

### 3.2. Clinical Characteristics of the Study Population

Compared to HC, T2DM and DN patients were found to have significantly higher levels of FBS, HbA1c, insulin, and insulin and homeostasis model assessment-estimated insulin resistance (HOMA-IR), indicating IR. In addition, renal function tests found that urea and creatinine were significantly elevated, whereas GFR (mL/min/1.73 m$^2$) was significantly lower in the DN cases, suggesting renal damage. T2DM and DN patients also had significantly higher levels of LDL (low-density lipoprotein), HDL (high-density lipoprotein), and triglycerides in their lipid profiles compared to HC.

### 3.3. Comparative Analyses of the mRNA Expression of Three Genes (PTEN, MMP2, and NF-κB) and MicroRNA-181b-p for T2DM, DN, and HC

The study population was assessed for the fold change (FC) of circulating PTEN, NF-κB, and MMP2 using RT-PCR. The FCE of PTEN was significantly downregulated in T2DM and DN patients compared to HC, with T2DM patients showing (0.84 $\pm$ 0.85) and DN patients showing (0.63 $\pm$ 0.64). On the other hand, NF-κB was significantly upregulated in T2DM and DN compared to HC, with T2DM patients showing (1.72 $\pm$ 1.29) and DN patients showing (1.80 $\pm$ 1.29). Similarly, circulating MMP2 was non-significantly downregulated in T2DM and DN compared to HC, with T2DM patients showing (0.84 $\pm$ 1.03) and DN patients showing (0.70 $\pm$ 0.55). The circulating hsa-miR-181b-5p was upregulated in T2DM

and DN compared to HC, with T2DM patients showing (2.09 ± 0.09) and DN patients showing (3.16 ± 2.79). (Tables S6 and S8, Figure 2).

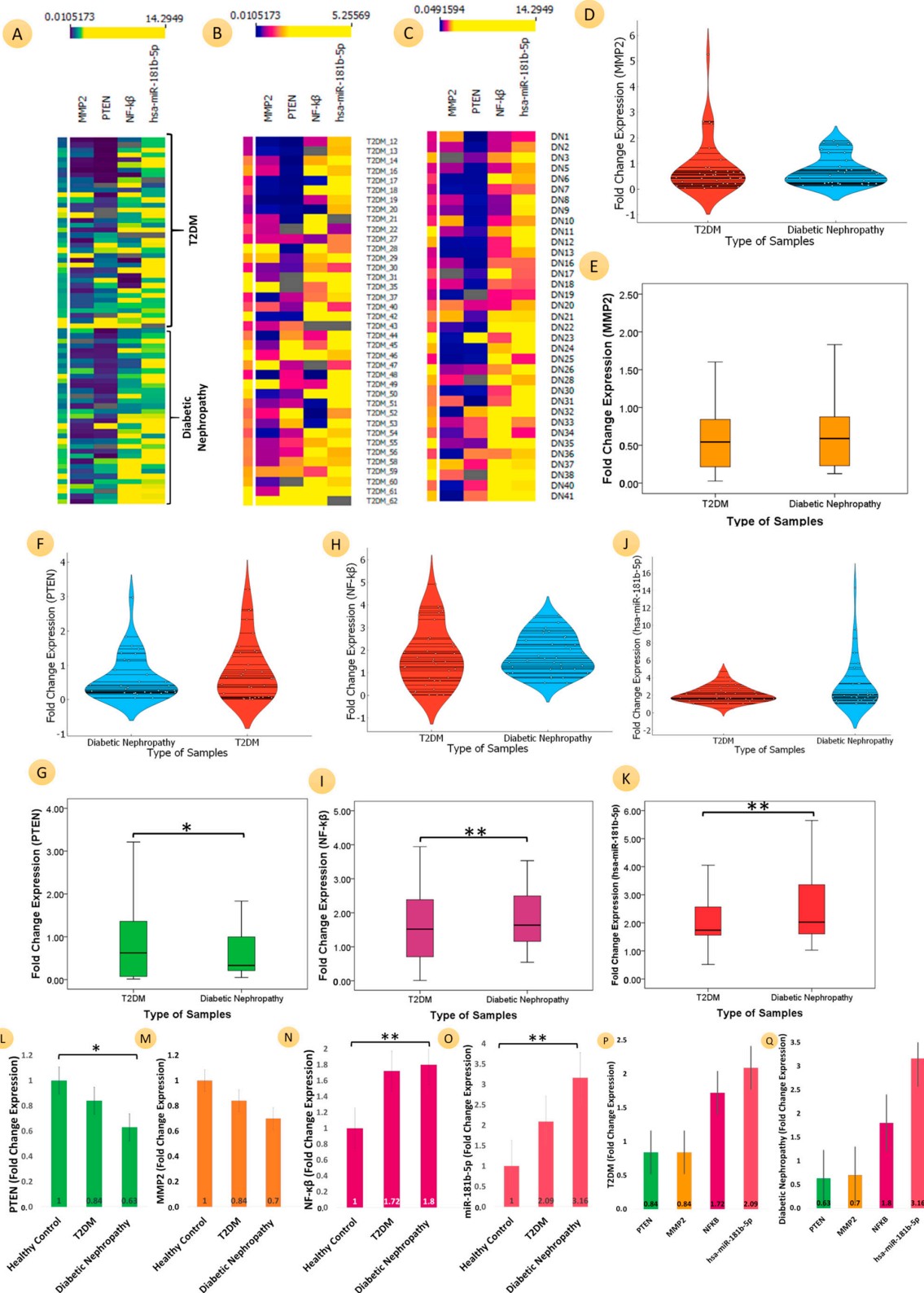

**Figure 2.** Comparative analyses of the mRNA expression of three genes (PTEN, MMP2, and NF-κB) and microRNA-181b-5p for T2DM patients, diabetic nephropathy patients, and healthy controls. Heat map (**A**) for both T2DM and diabetic nephropathy samples (**B**) for T2DM samples (**C**) for diabetic nephropathy

samples for three genes (PTEN, MMP2, and NF-κB) and microRNA-181b-5p (the fold change expression values represent the upper part of heat map with their respective colour densities); (**D**) violin plot showing the entire FC distribution of two groups for T2DM (brick red) and diabetic nephropathy (blue) for MMP2; The boxplots show the distribution of data (minimum, first quartile, median, third quartile, and maximum) for (**E**) MMP2 (yellow colour), T2DM (0.84 ± 1.05), and diabetic nephropathy (0.71 ± 0.56) (Student's t value: 0.708, *p*-value: 0.482); (**F**) violin plot of T2DM (brick red) and diabetic nephropathy (blue) for three genes PTEN. (**G**) PTEN (green colour), T2DM (0.84 ± 0.86), and diabetic nephropathy (0.63 ± 0.66) (Student's t value: 1.127, *p*-value: 0.264); (**H**) violin plot showing T2DM (brick red) and diabetic nephropathy (blue) for NF-κB; (**I**) NF-κB (dark pink colour), T2DM (1.72 ± 1.29), and diabetic nephropathy (1.81 ± 0.86) (Student's t value: 0.309, *p*-value: <0.758); (**J**) violin plot of T2DM (brick red) and diabetic nephropathy (blue) for microRNA-181b-5p. (**K**) hsa-miR-181b-5p (red colour), T2DM (2.09 ± 0.09), and diabetic nephropathy (3.16 ± 0.2.79) (Student's t value: 2.147, *p*-value: <0.03). Bar graphs provide visualizations of different categorical data: (**L**) fold change (FC) of PTEN (green colour) (*p*-value: <0.05), significantly downregulated in similar trends in T2DM (FC: 0.84) and DN (FC:0.63); (**M**) FC of MMP2 (yellow colour), downregulated in similar trends in T2DM (FC: 0.84) and DN (FC:0.7); (**N**) FC of NF-κB (pink colour) (*p*-value: <0.001), significantly upregulated in similar trends in T2DM (FC: 1.72) and DN (FC:0.8); (**O**) FC of hsa-miR-181b-5p (coral colour) (*p*-value: <0.001), significantly upregulated in similar trends in T2DM (FC: 2.09) and DN (FC:3.16); FC of different genes PTEN (FC: 0.84) and MMP2 (FC: 0.84) were downregulated, whereas the FC of NF-κB (FC: 1.72) was upregulated; similarly, hsa-miR-181b-5p was also upregulated in (**P**) T2DM and (**Q**) diabetic nephropathy. (** ≤ 0.05; * ≤ 0.001).

### 3.4. Comparative Analyses of the mRNA Expression of Three Genes (PTEN, MMP2, and NF-κB) and MicroRNA-181b-5p in Patients with Insulin Resistance

Comparative analyses were performed for the mRNA expression of three genes (PTEN, MMP2, and NF-κB) and of microRNA-181b-5p for two groups: insulin-sensitive (IS) (control group) and insulin resistance (IR) (Case Group). The classification criteria for the samples were based on HOMA-IR values (IS: ≤2.5; IR: >2.5) [18].

The PTEN expression in IR patients (0.70 ± 0.57) was significantly (*p* = 0.002) downregulated compared to patients who are sensitive to insulin (control group). The FC of PTEN in insulin-resistant T2DM (0.75 ± 0.62) and DN (0.66 ± 0.52) patients showed a similar trend of downregulation with the progression of the disease (Figure 3; Supplementary Tables S7 and S8). MMP2 expression among patients with IR (0.77 ± 0.57) was downregulated compared to patients who were IS (control group). The FC of MMP2 in IR-T2DM (0.83 ± 0.78) and DN (0.71 ± 0.58) showed a similar trend of downregulation with the progression of the disease. NF-κB expression in patients with IR (1.83 ± 1.02) was significantly (*p* = 0.000) upregulated compared to patients who were IS. However, the FC of NF-κB was not affected in IR T2DM (1.82 ± 1.67) and DN (1.84 ± 0.88) groups. hsa-miR-181b-5p expression in patients with IR (2.698 ± 2.30) was significantly (*p* = 0.000) upregulated compared with IS (control group). The FC of hsa-miR-181b-5p in IR T2DM (1.82 ± 1.67) and DN (1.84 ± 0.88) groups showed a similar trend of upregulation with the progression of the disease. Figure 3D presents a radar plot that shows the significant KEGG pathways influenced by the genes that were studied. The values on the plot represent the -log10 FDR (false discovery rate) of the pathways. Two important pathways, endocrine resistance (MMP2) and IR (PTEN, NF-κB), were found to be influenced by the studied genes. Several other pathways were also found to be influenced by the studied genes, including the T cell receptor signalling pathway (NFKBIA, NFKB1, CHUK, RELA, and PI3R1), the PI3K-Akt signalling pathway (NFKB1, TP53, CHUK, PTEN, RELA, and PIK3R1), Th1 and Th2 cell differentiation (NFKBIA, NFKB1, CHUK, and RELA), the IL signalling pathway (NFKBIA, NFKB1, CHUK, and RELA), the AGE-RAGE signalling pathway in diabetic complications (MMP2, NFKB1, RELA, and PIK3R1), endocrine resistance (MMP2, TP53, and PIK3R1), and the mTOR signalling pathway (CHUK, PTEN, and PIK3R1). Overall, these findings suggest that the studied genes play an important role in various cellular pathways and processes, including

IR, endocrine resistance, immune cell signalling, cellular growth and differentiation, and renal fibrosis.

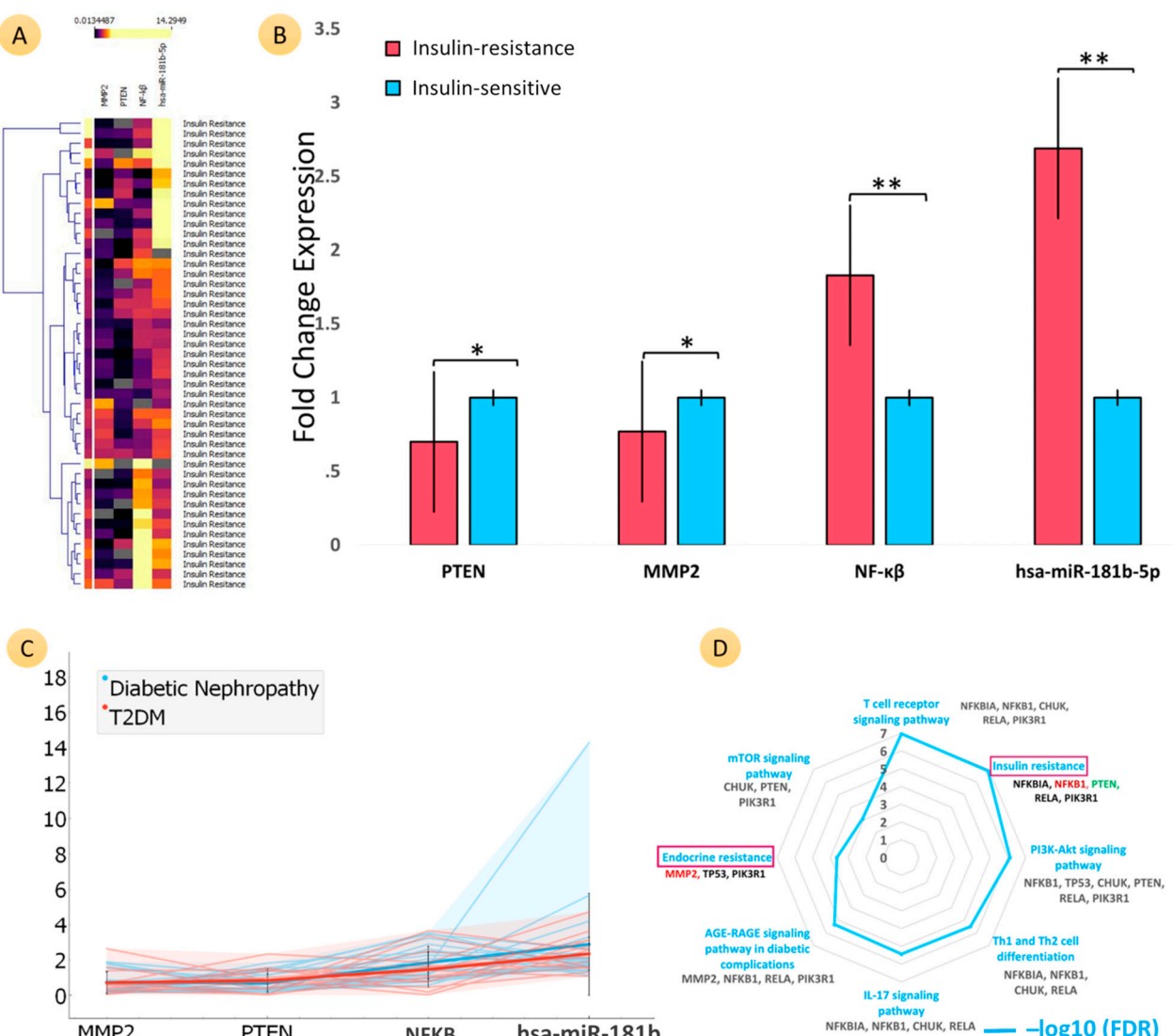

**Figure 3.** Comparative analyses of the mRNA expression of three genes (PTEN, MMP2, and NF-κB) and microRNA-181b-5p for two groups: insulin-sensitive (IS) (control group) and insulin resistance (IR) (case group). The classification criteria of the samples were based on HOMA-IR values (IS: ≤2.5; IR: >2.5). Heat map (**A**) shows the fold change (FC) of genes PTEN, MMP2, and NF-κB and of microRNA-181b-5p for both T2DM and diabetic nephropathy samples of IR patients (fold change expression (FCE) values represent the upper part of heat map with their respective colour densities). (**B**) Bar graphs show the comparative FCs of PTEN (IR FC: 0.70 ± 0.57), MMP2 (IR FC: 0.77 ± 0.69), and NF-κB (IR FC: 1.83 ± 1.02) as well as miR-181b-5p (IR FC: 2.69 ± 2.30) with the control group (IS FC: 1 ± 1.05). PTEN and MMP2 were significantly downregulated for IR patients, and NF-κB and hsa-miR-181b-5p were significantly upregulated. Bar plot colour representation: the insulin resistance group is the coral colour, and the insulin-sensitive group is the blue colour. ** ≤ 0.05; * ≤ 0.001. (**C**) Trend plot shows the complete visualization of FC distribution and the entire trend of all genes and microRNA for the IR groups of both T2DM and DN patients. The FCs of PTEN and MMP2 were more downregulated and NF-κB and hsa-miR-181b-5p were more upregulated in DN patients compared to T2DM without nephropathy for IR patients. (**D**) Radar plot (values: −log10 FDR) represents the significant KEGG pathways in two important pathways, endocrine resistance (MMP2) and IR (PTEN and NF-κB), influenced by our studied genes.

*3.5. Comparative Analyses of mRNA Expression of Three Genes (PTEN, MMP2, and NF-κB) and MicroRNA-181b-5p for Five Stages of Chronic Kidney Disease (CKD)*

The study population's GFR was calculated using the MDRD formula, and CKD was classified into five stages based on severity. The FC of MMP2 demonstrated a continuous trend of downregulation with the severity of CKD, and PTEN's FC also displayed a significant downregulation trend ($p = 0.05$). On the other hand, NF-κB expression exhibited a significant upregulation trend ($p = 0.05$) with CKD severity. The FC of hsa-miR-181b-5p showed a significant upregulation trend ($p = 0.05$) with renal damage. The trend plot presented in Figure 4E shows the complete visualization of the FC distributions and the trend of all genes and microRNAs in a single graph. The trend analysis reveals that PTEN and MMP2 were downregulated and NF-κB and hsa-miR-181b-5p were upregulated with the progression of CKD. The stages of CKD were categorized based on GFR and the percentage of kidney function, and the mean values for the FCs of the genes and microRNA were calculated for each stage. The FC of MMP2 remained relatively consistent across all stages, whereas PTEN and NF-κB's FCs decreased with the progression of the disease (Figure 4 and Table S11).

*3.6. Network Construction, Gene Ontology, and Interatomic Validation from Similar MicroRNA and Gene Expression Omnibus Datasets*

3.6.1. Construction of Protein–Protein Interaction Network with PTEN, NF-κB, and MMP2 and Interacting Genes

The PTEN, NF-κB, and MMP2 and their interactions with nine other genes were used to establish the PPI network with STRING, which constituted 13 nodes, 35 edges, and a PPI enrichment $p$-value $< 0.002$ at medium confidence (0.400). (Figure 5A) All 13 genes were involved in PPI networks. Among these, PTEN was directly connected with SLC9A3R1, NEDD4, MMP2, TP53, NF-κBIA, CHUK, and PIK3R1, whereas MMP2 was directly connected with TIMP1, NF-κBIA, TIMP2, PTEN, and TP53, and NF-κB1 directly interacted with NF-κBIA, TP53, RELB, RELA, CHUK, and PIK3R1. The aforementioned interatomic combinations were involved in many regulatory metabolic pathways and in the gene ontology of the body.

3.6.2. KEGG Pathway Enrichment Analysis

The genes involved in the construction of PPI networks were explored in the KEGG pathways and were observed to be involved with T cell receptors, IR, PI3K-Akt signalling, Th1 and Th2 cell differentiation, IL-17 signalling, AGE-RAGE signalling pathways in diabetic complications, endocrine resistance, and mTOR signalling (Figure 5B and Table S12).

3.6.3. Gene Ontology Analysis

The gene ontology analysis performed for the molecular characterization of those genes involved in the PPI network resulted in the biological process (BP) and molecular function (MF) categories (Figure 5C,D and Tables S11 and S12).

3.6.4. Validation of Gene Expression Profiles from GEO Datasets

We found three datasets that compared the circulating gene expression between T2DM and HC. We searched out, in T2DM and HC samples, the individual microarray expression of all genes involved in PPI networks. The heat maps in Figure 5E–G represent differential expression intensities of individual samples. The FCs of every gene are represented in a bar plot, with MMP2 ($p = 0.01$; downregulated), PTEN ($p = 0.02$; downregulated), and NF-κB ($p = 0.02$; upregulated) showing significant changes. This was in concordance with our wet lab expression profile. Apart from this, TP-53 ($p = 0.00$) and NEDD4 ($p = 0.00$) also showed significant alterations in their expressions in the microarray studies (Figure 5H and Table S15).

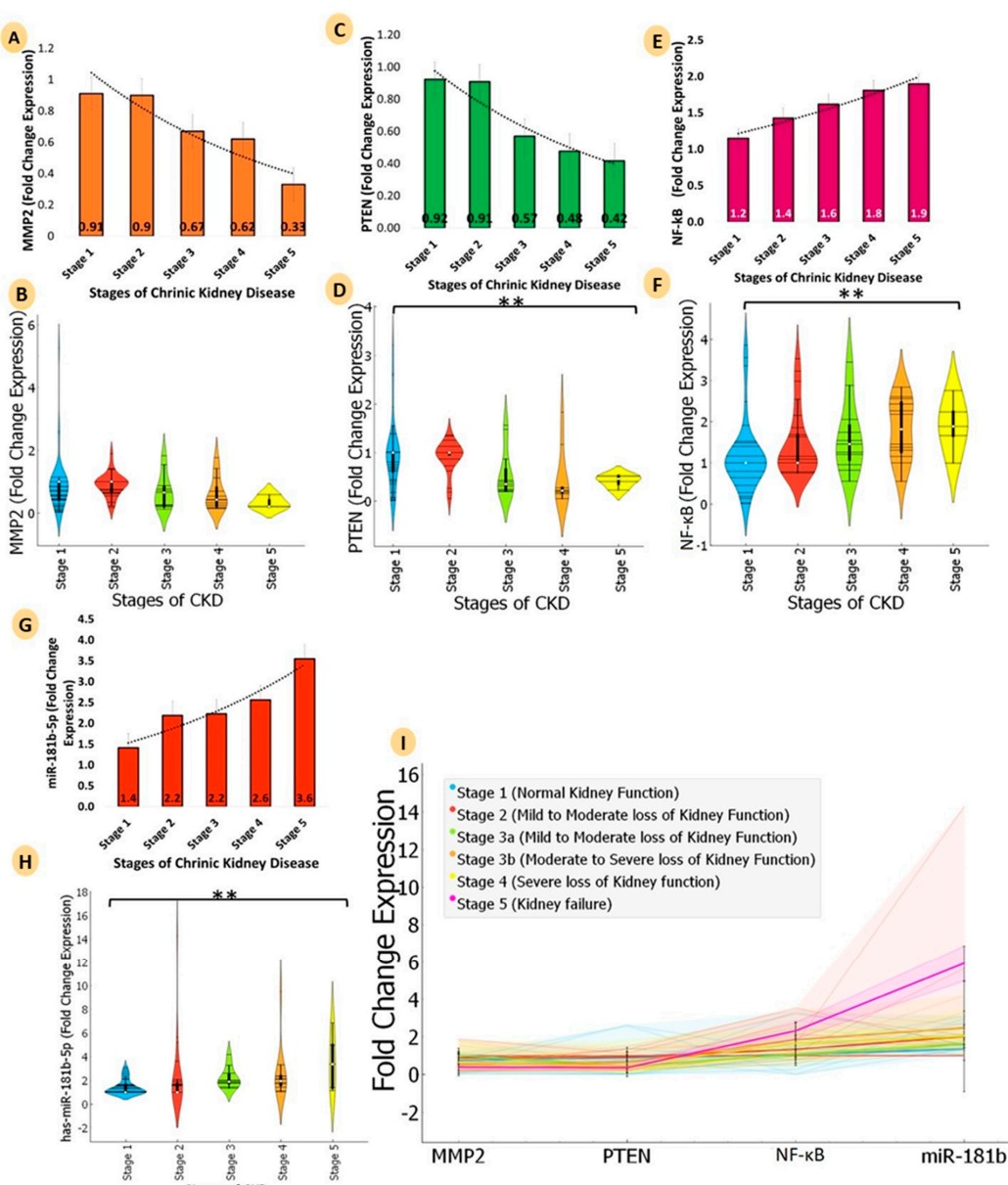

**Figure 4.** Comparative analyse the mRNA expression of three genes (PTEN, MMP2, and NF-κB) and microRNA-181b-5p for five stages of chronic kidney disease (stage 1: normal kidney function; GFR: 90 or higher; % of kidney function: 90–100%; stage 2: kidney damage with mild loss of kidney function; GFR: 89–60; % of kidney function: 89–60%; stage 3: (3a: kidney damage with mild to moderate loss of kidney function; 3b: moderate to severe loss of kidney function); GFR: 59–30; % of kidney function: 59–30%; stage 4: severe loss of kidney function; GFR: 29–15; % of kidney function: 29–15%; stage 5: kidney failure; GFR: less than 15; % of kidney function: less than 15%). Bar graphs provide a visualization of FCs of chronic kidney disease at different stages. (**A**) MMP2 (yellow colour) showed a continuous downregulation trend (stage 1, FC: 0.91; stage 2, FC: 0.91; stage 3, FC: 0.67; stage 4, FC: 0.62; stage 5, FC: 0.33). (**B**) The violin plot shows the entire FC distribution of the five stages of CKD for MMP2 (**C**) Bar graphs for PTEN (green colour) showed a continuous significant downregulation trend (stage 1, FC: 0.92; stage 2, FC: 0.91; stage 3, FC: 0.57; stage 4, FC: 0.48; stage 5, FC: 0.42). (**D**) The violin plot shows the entire FC distribution of the five stages of CKD for PTEN (**E**) NF-κB (pink colour) showed a continuous significant upregulation trend (stage 1, FC: 1.2; stage 2, FC: 1.4; stage 3, FC: 1.6; stage 4, FC: 1.8; stage 5, FC: 1.9). (**F**) FC distribution of the five stages of CKD for NF-κB (**G**) hsa-miR-181b-5p (red colour) showed a continuous significant upregulation trend (stage 1, FC: 1.4; stage 2, FC: 2.2; stage 3, FC: 2.2; stage 4, FC: 2.6; stage 5, FC: 3.6). (**H**) The Violin plot

shows the entire FC distribution of the five stages of CKD microRNA-181b-5p, and (**I**) shows the trend plot visualization of FC distribution and of the entire trend of all genes and microRNA in a single graph. Violin plot colour representation: stage 1, sky blue; stage 2, brick red; stage 3, lawn green; stage 4, orange; stage 5, yellow. ** $\leq 0.05$.

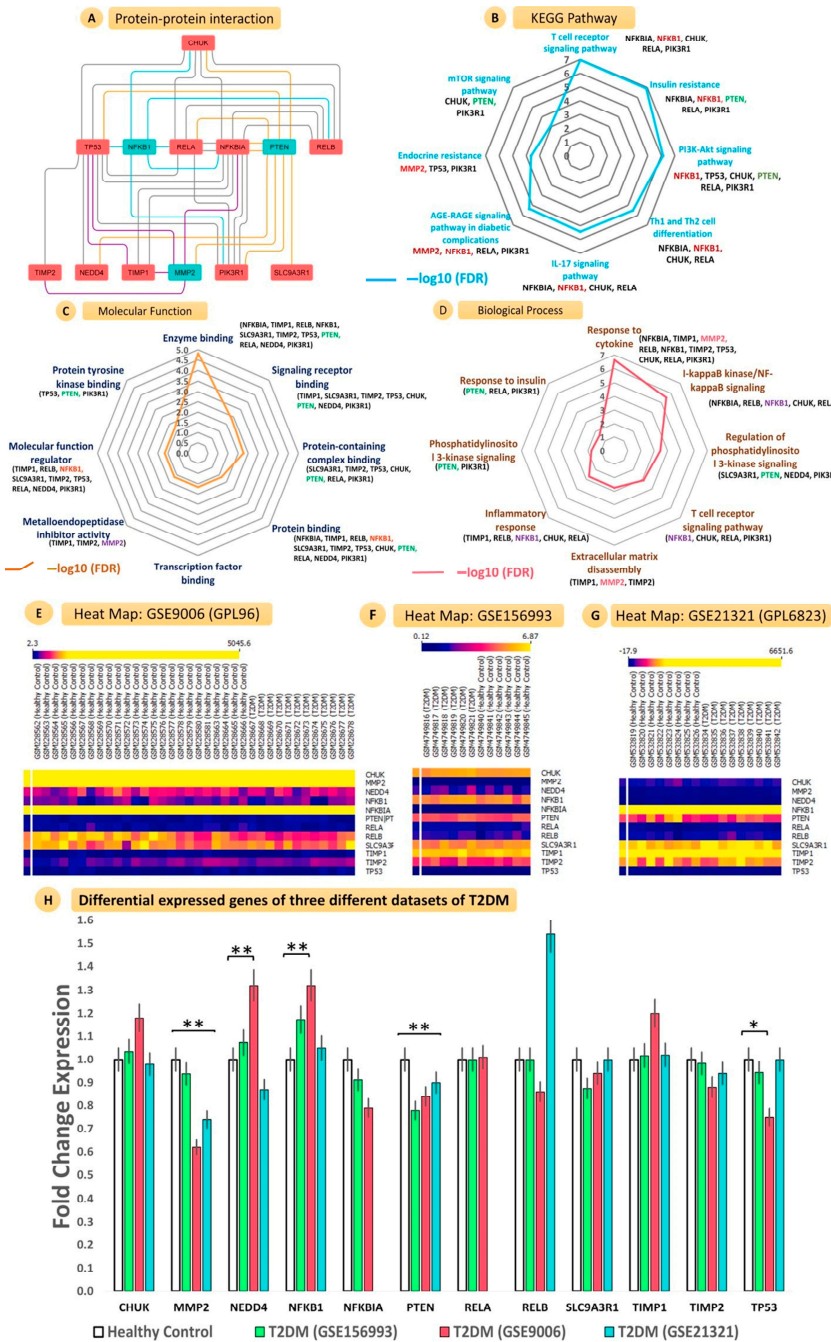

**Figure 5.** Interatomic connections, gene ontology, pathway enrichment, and validations using gene expression omnibus microarray datasets. The protein–protein interaction diagram (**A**) shows the strong interactions of PTEN, MMP2, and NF-κB with ten other proteins, CHUK, PIK3R1, RELA, NF-κBIA, TIMP1, SLC9A3R1, TIMP2, NEDD4, TP53, and RELB. (**F**) Radar plot (values: −log10 FDR) represents the significant (**B**) KEGG pathways, (**C**) molecular functions, and (**D**) biological processes regulated by PPI-interacting genes. Heatmap shows the microarray expression of individual samples for the (**E**) GSE9006 (GPL96), (**F**) GSE156993, and (**G**) 21321 (GPL6823) datasets. (**H**) Bar graphs show the comparative FCs of all PPI genes for three PBMC T2DM datasets (** $\leq 0.05$; * $\leq 0.001$).

### 3.6.5. Target Prediction of hsa-miR-181b-5p and Construction of miRNA Regulatory Networks

To identify the regulatory relationship of investigated genes and hsa-miR-181b-5p, four different target prediction databases were used for hsa-miR-181b-5p. PTEN and TIMP2 were observed to be the direct targets of hsa-miR-181b-5p. The co-expression networks of PTEN, NF-κB, and MMP2 and of ten other genes, viz. SLC9A3R1, NEDD4, PI3KR1, TP53, TIMP1, TIMP2, NF-κBIA, CHUK, RELA, and RELB, were constructed by using Cytoscape. We searched the similarly expressed circulating and kidney tissue gene expression microarray datasets of T2DM and compared them to the HC samples (Figure 6A).

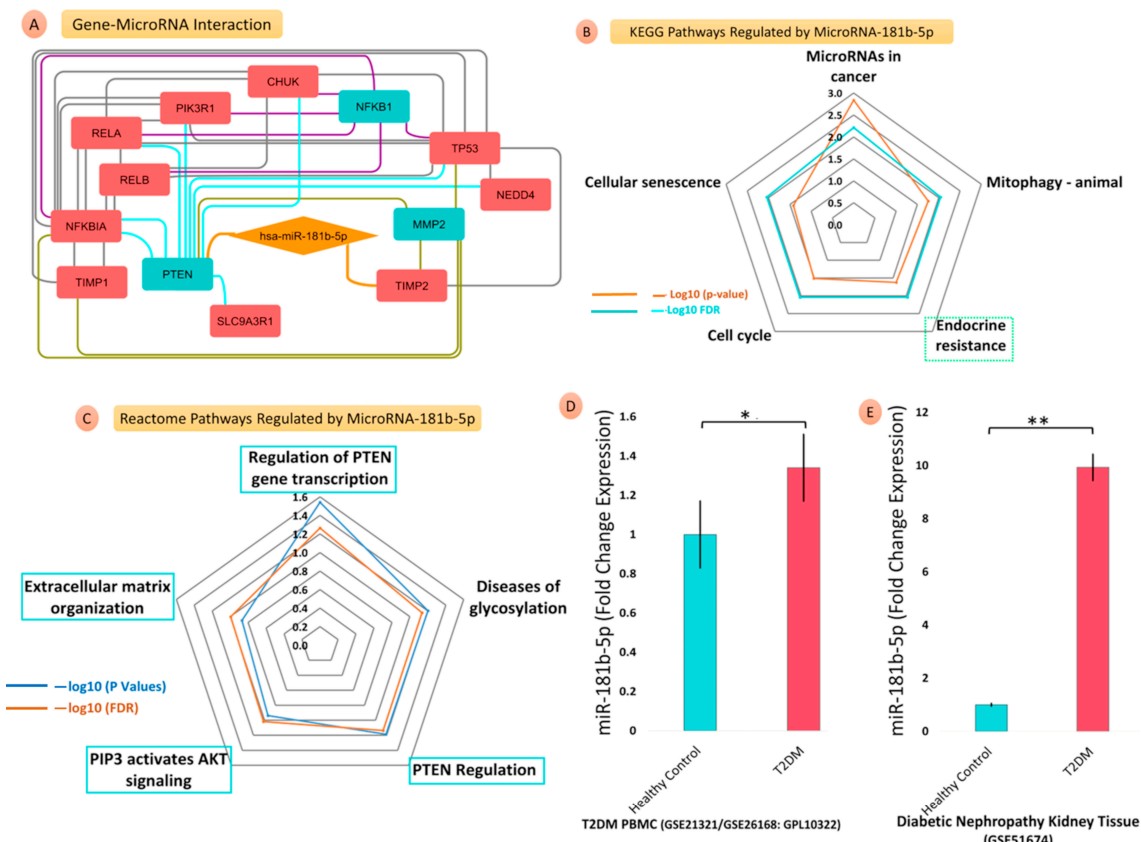

**Figure 6.** Gene–microRNA interaction, microRNA-regulated pathway enrichment, and validations using GEO microRNA–microarray datasets. The gene–microRNA interaction diagram (**A**) shows the direct target genes, PTEN and TIMP2, of hsa-miR-181b-5p; the radar plot (values: −log10 FDR) represents the significant (**B**) KEGG pathways and (**C**) reactome pathways regulated by miR-181b-5p; the bar graphs show the comparative FCs of miR-181b-5p for T2DM, PBMC, and DN kidney tissues datasets. Bar graphs show the comparative FCs of miR-181b-5p in (**D**) PBMC of T2DM datasets (GSE21321/GSE26168: GLP10322) and (**E**) Diabetic nephropathy kidney tissue datasets (GSE51674) (** $\leq$ 0.05; * $\leq$ 0.001).

### 3.6.6. Regulation of hsa-miR-181b-5p for KEGG and Reactome Pathway Enrichment

hsa-miR-181b-5p significantly regulated endocrine resistance, cell cycle, mitophagy-animal, and microRNAs in cancer and cellular senescence pathways. Similarly, hsa-miR-181b-5p was also involved in the regulation of PTEN gene transcription, PTEN regulation, PIP3's activation of AKT signalling, and the extracellular matrix (ECM) organization pathways of the reactome database (Figure 6B,C).

### 3.6.7. Validation of hsa-miR-181b-5p Expression Profile from GEO Datasets

We found two datasets that compared the circulating gene expressions between T2DM (Blood), DN (Tissue), and HC. We searched out, in T2DM, DN and HC samples, the individual microarray expression of hsa-miR-181b-5p involved in the microRNA–protein

network. The FC of hsa-miR-181b-5p, which is represented in a bar plot, showed significant alterations in expression in the microarray studies. (Figure 6D,E and Tables S16 and S17).

## 4. Discussion

PTEN is a phosphatase encoded by the PTEN gene that negatively regulates protein kinase B (Akt) and mammalian target of rapamycin (mTOR) pathways. Important characteristics of DN are renal tubulointerstitial fibrosis (TIF) and epithelial–mesenchymal transition (EMT), and PTEN plays a significant role in the regulation of both. Regulation of the PI3-kinase/AKT pathway by PTEN is attributed to its effects on reconverting phosphatidylinositol 3,4,5-triphosphate (PIP3) back to phosphatidylinositol 3,4-bisphosphate (PIP2) [6]. PI-3 kinase (PI3K) is a crucial molecule in the insulin signalling pathway and its downstream effect on GLUT4 translocation and subsequent glucose uptake. Further, PI3K activates mTOR, which promotes protein synthesis, nutrient uptake, and increased glycolysis. Regulation of the action of insulin is brought about by PTEN, PI3K, and its effects on the phosphorylation and dephosphorylation of intermediaries [6]. In the current study, PTEN was significantly downregulated in T2DM and DN patients when compared to HC. The PTEN expression in patients with IR was significantly downregulated compared to HC. Furthermore, the expression of PTEN in T2DM and DN patients showed a similar trend of downregulation with the progression of disease (Figure 7). The FC of PTEN showed consistent downregulated with progression of renal damage, thereby suggesting that an enhanced renal TIF may be triggered and sustained due to the alleviation of the negative control of PTEN.

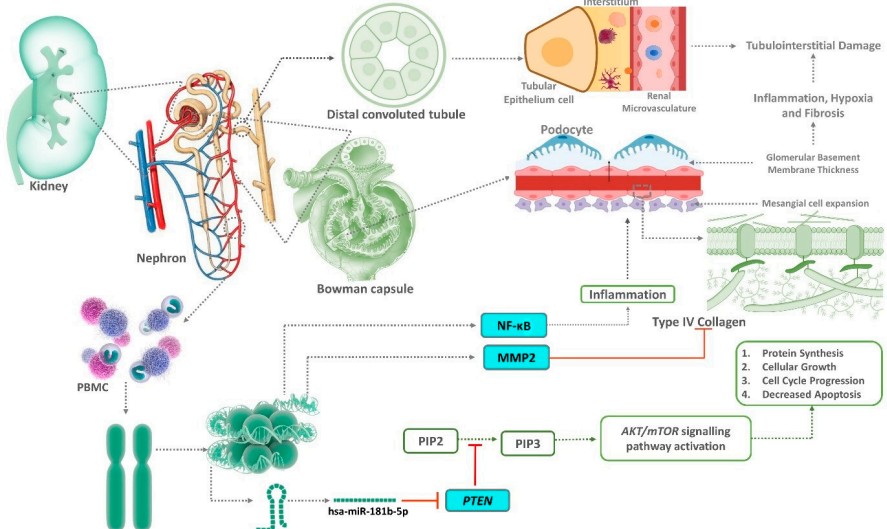

**Figure 7.** Role of circulating PTEN, MMP2, and NF-κB as well as their target microRNA-181b in renal fibrosis via the AKT/mTOR signalling pathway during aggravation of diabetic nephropathy. PTEN antagonizes the PI3-kinase/AKT pathway by reconverting phosphatidylinositol 3,4,5-triphosphate (PIP3) back to phosphatidylinositol 3,4-bisphosphate (PIP2). The activation of mammalian target of rapamycin (mTOR) is responsible for diabetic nephropathy. Hyperglycaemia stimulates phosphatidylinositol-3 kinase (PI3K) and protein kinase B(AKT) pathways, which subsequently activates mTOR. Downregulated MMP2 and activated mTOR is responsible for basement membrane thickening, mesangial matrix expansion, and renal fibrosis. NF-κB caused an inflammatory response with the progression of the disease.

In the current study, our search in the miRnet, miRDB, and TargetScan databases yielded hsa-miR-181b-5p as an important microRNA targeting PTEN. miR-181b is described as an oncogenic miRNA in cancers [19]. In addition to its oncogenicity, miR-181b was also demonstrated to have a crucial role in liver fibrosis. PTEN, which has a protective role in liver fibrosis, was shown to be targeted by miR-181b in causing liver fibrosis progression [20]. This regulation of PTEN by miR-181 occurs via activation of the PI3K-

mTOR pathway and culminates in an accelerated response to growth factor signalling, resulting in the promotion of nutrient uptake, protein synthesis, and increased glycolysis. Contrastingly, proliferating thymocytes were demonstrated to have an increased miR-181 expression level that post-transcriptionally represses PTEN expression in response to increased metabolic demands. The net effect is the enhanced ability of the cells to meet the metabolic requirements for survival, growth, and proliferation [21]. In miR-181-deficient mice, an increase in PTEN activity leads to inhibition of PI3K-mTOR pathway and an inability of cells to make necessary metabolic adjustments, resulting in the disruption of thymocyte development, a block in NKT cell development, and suppression of innate immunity [22].

Aberrant expression of microRNA (miRs) results in renal complications of diabetes, including renal hypertrophy and matrix protein accumulation. The reduced expression of PTEN in hyperglycaemia accentuates these processes. Altered expressions of miR were demonstrated to downregulate expression of PTEN [23]. miR-21 promotes renal fibrosis in DN by targeting PTEN and SMAD7 [24]. In the current study, hsa-miR-181b-5p was observed to be upregulated in patients with IR, T2DM, and DN patients. We demonstrated hsa-miR-181b-5p to be significantly increased with the severity of DN in association with a concurrent decrease in its direct target, PTEN.

Injury to the kidney at early stages of CKD leads to the secretion of various pro-inflammatory and pro-fibrotic cytokines, which drives the progression of renal interstitial fibrosis. The activity of matrix metalloproteinases such as MMP2 and MMP9 are altered during injury to a renal basement membrane-promoting phenotype transformation of renal tubular epithelial cells, resulting in the excessive deposition of ECM. In advanced stages of CKD, the activity of MMP2 and MMP9 is decreased, which leads to the inadequate degradation of ECM and, subsequently, fibrosis. MMP2, in CKD, was also correlated with proteinuria and intima–media thickness [25]. The decreased activity of MMP2 in the advanced stage of CKD is attributed to the enhanced endocytosis processes [26]. A high-fat diet can lead to a state of chronic inflammation that affects ECM synthesis and degradation. This can lead to the development of IR [27]. Interestingly, IR in animal models was associated with a decreased level of MMP2 in adipose tissue [28]. Our finding is in concordance with the observations of Lewandowski et al., who demonstrated decreased MMP2 blood levels in patients with T2DM when compared to nondiabetic controls [29], thereby leading to an accumulation of ECM and basement membrane thickening. Further, our study observed a decrease in FC of MMP2 in patients with increased IR resulting from T2DM and DN (Figure 3B). We also observed a decreasing trend in MMP2 expression as DN progressed from early stages to the advanced stage (Figure 2D,E).

NF-κB is a family of transcription factors that control production of proinflammatory proteins, and it was demonstrated that it plays a key role in the pathogenesis of various complications of diabetes. Persistent hyperglycaemia activates NF-κB, which triggers the expression of various cytokines, chemokines, and cell adhesion molecules. Its overexpression also triggers the calcification of endothelial cells, thereby leading to endothelial dysfunction [30]. In addition, Austin et al. suggested that increased NF-κB signalling may be involved in the pathogenesis of IR [31]. The activation of the NF-κB pathway precedes the development of glomerular lesions in diabetes. Hence, the NF-κB system is considered an important therapeutic target against the progression of human DKD [32]. Interestingly, PTEN phosphatase dephosphorylates NF-κB-activating protein (NKAP) and limits NFκB activation to suppress the expression of PDHK1, a NF-κB target gene. Experimental inhibition of NF-κB prevented renal oxidative stress, glomerular inflammation, and injury. Our study demonstrates NF-κB is upregulated in patients with DN when compared to diabetic patients without DN. We also observed that NF-κB was upregulated in patients with IR patients. Furthermore, our data depict an increasing trend in the expression of NF-κB with the progression of DN (Figure 2H,I).

PTEN has an inhibitory effect on the PI3-kinase/AKT pathway due to its effects that convert PIP3 to PIP2. Therefore, decreased PTEN expression would lead to increased activation of the PI3-kinase/AKT pathway and its downstream targets, such as the mammalian

target of rapamycin (mTOR). An activated AKT/mTOR pathway due to hyperglycaemia was attributed as one of the reasons for the development of DN [6]. Hyperglycaemia stimulates PI3K/AKT pathways, causing the subsequent activation of mTOR. An activated mTOR with a concurrent decreased expression of MMP2 leads to the inhibition of autophagy, basement membrane thickening, mesangial matrix expansion, and renal fibrosis as observed in DN [6]. In addition, an increased expression of NF-κB accentuates the process by aggravating the inflammatory response, leading to the progression of T2DM to DN (Figure 7).

Our in-silico analysis identified the endocrine resistance (MMP2) and IR (PTEN and NF-κB) pathways as two important KEGG pathways influenced by our genes of interest (Figure 4D). The altered expression of the genes (PTEN, MMP2, and NF-κB) and microRNA (hsa-miR-181b-5p) observed in our study in patients with IR highlights the possible association of these genes as leading to IR in diabetic patients (Figure 8).

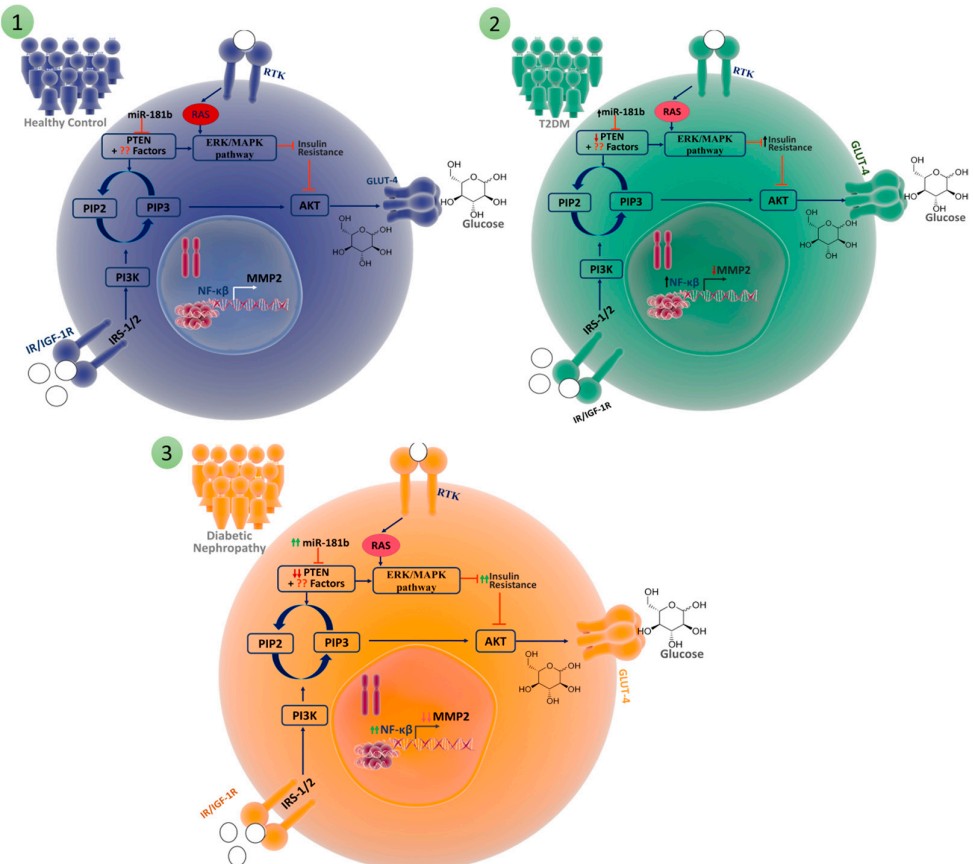

**Figure 8.** We have illustrated the pathway in three distinct conditions: (**1**) healthy controls, (**2**) individuals with type 2 diabetes, and (**3**) individuals with diabetic nephropathy. These conditions were compared to analyze insulin resistance and its impact on the pathway with diabetic nephropathy for comparative analysis via the collective effect of the balance between the ERK/MAPK pathway and the PTEN-Akt-m-TOR pathway. PI3K plays a crucial role in the insulin signalling pathway and on GLUT4 translocation, affecting the response of the cell to hyperglycaemia. PTEN antagonizes the PI3-kinase/AKT pathway by reconverting phosphatidylinositol 3,4,5-triphosphate (PIP3) back to phosphatidylinositol 3,4-bisphosphate (PIP2). PTEN acts as a negative regulator of the PI3K/Akt pathway and as a positive regulator of the ERK/MAPK pathway. Hence, the phenotype of IR is a collective effect of the balance between the two pathways (PI3K/Akt and ERK/MAPK) and is regulated by multiple factors, including PTEN [6,7]. Therefore, the altered PTEN expression observed in our study in patients with DN, in conjunction with other key regulators, may have tilted the balance towards IR, facilitating the progression of T2DM to DN.

Therefore, the altered PTEN expression observed in our study in patients with DN, in conjunction with other key regulators, may have tilted the balance towards IR, facilitating the progression of T2DM to DN [33]. Further studies are required to explore and identify these key regulators that work in tandem with PTEN to tilt the balance between the PI3K/Akt and ERK/MAPK pathways towards IR in patients with DN.

## 5. Conclusions

In patients with T2DM, when compared to HC, the expression of genes MMP2 and PTEN was decreased, whereas NF-κB and microRNA-181b-5p were increased. This trend was observed to worsen in patients with DN when compared to those with T2DM alone. KEGG pathway analysis identified endocrine resistance (MMP2) and insulin resistance (PTEN, NF-κB) as the two pathways modulated by the genes of interest. The progressive insult in patients with T2DM aggravates the derangement of the aforementioned pathways, culminating in the development of DN.

**Supplementary Materials:** The following supporting information can be downloaded at: https://www.mdpi.com/article/10.3390/kidneydial3010011/s1, Table S1. Description of Gene Expression Omnibus datasets of T2DM and healthy controls; Table S2. Description of selection criteria for sorting of Gene Expression Omnibus datasets of T2DM and healthy controls; Table S3. Description of microRNA Expression Omnibus datasets of T2DM and healthy controls; Table S4. Description of selection criteria for sorting of microRNA Expression Omnibus datasets of T2DM and healthy controls; Table S5. ANOVA test for clinical, biochemical, and gene and microRNA expression profiles for T2DM patients, DN patients, and healthy control subjects; Table S6. ANOVA test for circulating genes (PTEN, MMP2, and NF-κB) and microRNA (hsa-miR-181b-5p) expression profiles for T2DM patients, DN patients, and healthy control subjects; Table S7. T-test for circulating genes (PTEN, MMP2, and NF-κB) and microRNA (hsa-miR-181b-5p) expression profiles for T2DM patients, DN patients, and healthy control subjects; Table S8. Multinomial logistic regression analysis of circulating genes (PTEN, MMP2, and NF-κB) and microRNA (hsa-miR-181b-5p) expression profiles for T2DM patients, DN patients, and healthy control subjects; Table S9. T-test for circulating genes (PTEN, MMP2, and NF-κB) and microRNA (hsa-miR-181b-5p) expression profiles for insulin resistance patients and insulin-sensitive subjects; Table S10. ANOVA test for circulating genes (PTEN, MMP2, and NF-κB) and microRNA (hsa-miR-181b-5p) expression profiles for T2DM (Insulin Resistance), and DN (Insulin Resistance) patients and healthy control (Insulin Sensitive) subjects; Table S11. ANOVA test for circulating genes (PTEN, MMP2, and NF-κB) and microRNA (hsa-miR-181b-5p) expression profiles for five stages of chronic kidney disease; Table S12. Pathway enrichment (KEGG pathways) analysis for PPI genes using STRING database; Table S13. Gene ontology (biological process) analysis for PPI genes using STRING database; Table S14. Gene ontology (molecular functions) analysis for PPI genes using STRING database; Table S15. Validation of fold change expression for PPI interactions and key genes in three different microarray datasets of circulating T2DM and healthy subjects; Table S16. Validation of fold change expression for hsa-miR-181b-5p in three different microarray datasets (GSE21321) of circulating T2DM and healthy subjects; Table S17. Validation of fold change expression for hsa-miR-181b-5p in three different microarray datasets (GSE51674) of kidney tissue of T2DM and healthy subjects.

**Author Contributions:** Concept and design: M.K. and P.P.; data acquisition, analysis, and interpretation: M.K., S.T. and P.P.; manuscript drafting: M.K., S.T., P.P., A.G., R.G.A., N.K.B., G.K.B. and R.K.S.; manuscript revision: M.K., S.T. and P.P.; project supervision: P.P. All authors have read and agreed to the published version of the manuscript.

**Funding:** This work was supported by the Research Society for the Study of Diabetes in India "RSSDI/HQ/Grants/2022/02".

**Institutional Review Board Statement:** This study was carried out following the principles of the Declaration of Helsinki for medical research involving human subjects, and ethical approval was obtained from the Institutional Ethics Committee of AIIMS, Jodhpur ("Certificate Reference number: AIIMS/IEC/2019-20/792").

**Informed Consent Statement:** For all participants, freely given informed consent was obtained before inclusion in the study.

**Data Availability Statement:** The data presented in this study are available in the Supplementary Materials.

**Acknowledgments:** The authors are grateful to the All India Institute of Medical Sciences Jodhpur for providing the research facility to perform this in silico experiment. Manoj Khokhar is supported by a senior research fellowship of The University Grants Commission of India (NOV2017-361200).

**Conflicts of Interest:** The authors declare no conflict of interest.

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
