# Peer review of "PTEN, MMP2, and NF-κB and Regulating MicroRNA-181 Aggravate Insulin Resistance and Progression of Diabetic Nephropathy: A Case-Control Study"

_kidneydial, doi:10.3390/kidneydial3010011_

Round 1
Reviewer 1 Report
The aim of this study is to highlight the role of insulin resistance (IR) in the progression of nephropathy in T2DM via the PTEN-Akt-mTOR signaling pathway. Actually, the current proposal is interesting. As a result, I recommend that the current study be published after major revision using wet experiments to confirm the findings of this study.
Author Response
Reviewer 1
Comments and Suggestions for Authors
The aim of this study is to highlight the role of insulin resistance (IR) in the progression of nephropathy in T2DM via the PTEN-Akt-mTOR signaling pathway. Actually, the current proposal is interesting. As a result, I recommend that the current study be published after major revision using wet experiments to confirm the findings of this study.
Author Response: We were pleased to hear that you found our proposal interesting and appreciated your recommendation for publication. We understood your concern regarding the need for wet experiments to confirm our findings, and we agreed that this would be an important step towards validating our results.
In response to your suggestion, it is our humble submission that to confirm our bioinformatic results regarding Insulin resistance in diabetic nephropathy via PTEN-Akt-mTOR and its regulating miRNAs, we have performed the wet lab experiment using patient samples gene expression and miRNA expression profile. We have performed the gene expression of PTEN and the master regulator of inflammation NF-kB, which is under the control of PTEN indirectly.
Thank you again for your time and constructive feedback. We were committed to ensuring the quality and rigor of our research and strived to address any concerns or criticisms raised by reviewers in the best way possible.
Reviewer 2 Report
The manuscript title: “PTEN, MMP2 and NF-κB and regulating MicroRNA-181 aggravate insulin resistance and progression of Diabetic Nephropathy: Case-control study”, it is an interesting manuscript. They found the participation of PTEN, MMP2 and NF-κB in diabetic nephropathy patients and insulin resistance. The manuscript presents a lot of result that sustain their conclusions.
I have some comments that can improve their manuscript:
1) The authors need to define all the abbreviation and add a list of that. Some of the abbreviations are missing (abstract).
2) In foot graph the authors should put the name of groups equal than the plot (Figure 3)
Author Response
Reviewer 2
The manuscript title: “PTEN, MMP2 and NF-κB and regulating MicroRNA-181 aggravate insulin resistance and progression of Diabetic Nephropathy: Case-control study”, is an interesting manuscript. They found the participation of PTEN, MMP2, and NF-κB in diabetic nephropathy patients and insulin resistance. The manuscript presents a lot of results that sustain their conclusions.
Response: Thank you again for your feedback and for taking the time to review our manuscript. We appreciate your support and encouragement.
I have some comments that can improve their manuscript:
1) The authors need to define all the abbreviation and add a list of that. Some of the abbreviations are missing (abstract).
Response: We thank you for your review and feedback on our manuscript. We appreciate your suggestion to define all abbreviations and include a list of them to improve readability. We apologize for any confusion caused by missing abbreviations in the abstract. We have added a comprehensive list of abbreviations used in our study and defined them when first used in the text. Thank you for helping us improve the clarity of our manuscript.
2) In foot graph the authors should put the name of groups equal than the plot (Figure 3)
Response: Thank you for your review and feedback on our manuscript. We appreciate your suggestion regarding the labeling of the groups in the foot graph in Figure 3. In response to your suggestion, we have made sure to label the groups in the foot graph to match the plot in Figure 3. Thank you for helping us improve the accuracy of our figures. Once again, we appreciate your constructive feedback and thank you for taking the time to review our manuscript.
Reviewer 3 Report
The main achievement claimed in this manuscript is the demonstration of the progression of T2DM to DN by the authors. This was achieved by carefully monitoring the clinic and biological assessment. The authors specify the role of insulin resistance (IR) in the progression of nephropathy in T2DM via the PTEN-Akt-mTOR signaling pathway. The expression of MMP2, PTEN, NF-κB, and miR-181b-5p had been correlated with the progression of nephropathy to the severe stage.
However, I still have minor comments
-In the materials and methods, 2.5- Pathway analysis and...., needs more explanation to be understood.
- Please, make a little summary by linking healthy control to T2DM in 2 to 3 sentences before making your conclusion.
Author Response
Reviewer 3
The main achievement claimed in this manuscript is the demonstration of the progression of T2DM to DN by the authors. This was achieved by carefully monitoring the clinic and biological assessment. The authors specify the role of insulin resistance (IR) in the progression of nephropathy in T2DM via the PTEN-Akt-mTOR signaling pathway. The expression of MMP2, PTEN, NF-κB, and miR-181b-5p had been correlated with the progression of nephropathy to the severe stage.
Response: Thank you for reviewing our manuscript. We appreciated your feedback.
However, I still have minor comments
-In the materials and methods, 2.5- Pathway analysis and...., needs more explanation to be understood.
Response: Thank you for reviewing our manuscript and for your careful attention to detail. We appreciate your feedback and understood your concern regarding the clarity of our methodology section, particularly in section 2.5 on pathway analysis.
“Gene Ontology (GO) Enrichment is a method used to identify the functional categories of genes and their products that are over-represented in a set of differentially expressed genes. GO categorizes genes into three main categories: cellular component, biological process, and molecular function. Enrichment analysis can help researchers understand the biological functions and processes that are affected by a specific set of genes. In this study, Pathway Analysis and GO Enrichment were performed using STRING (version 11.5) to identify the biological pathways and networks associated with the genes of interest (PTEN, NF-ĸB, MMP2, and their interacting proteins). The significant Pathway Analysis results were ranked by the false discovery rate (FDR). Similarly, GO Enrichment was performed for cellular components (CC), biological processes (BP), and molecular functions (MF) related to the genes of interest. These analyses can provide insight into the underlying biological mechanisms and functions of the genes in the context of chronic kidney disease. in improving the clarity of our manuscript.”
- Please, make a little summary by linking healthy control to T2DM in 2 to 3 sentences before making your conclusion.
Response:
Thank you for reviewing our manuscript and providing us with valuable feedback. We appreciated your careful attention to detail and your constructive review of our conclusion section.
“In patients with T2DM when compared to HC, the expression of genes MMP2 and PTEN was decreased, while NF-ĸB and MicroRNA-181b-5p were increased. This trend was observed to worsen in patients with DN when compared to those with T2DM alone. KEGG pathway analysis identified endocrine resistance (MMP2) and insulin resistance (PTEN, NF-κB) as the two pathways modulated by the genes of interest. The progressive insult in patients with T2DM aggravates the derangement of the aforementioned pathways culminating in the development of DN.”

Reviewer 4 Report
In the manuscript entitled PTEN, MMP2 and NF-κB and regulating MicroRNA-181 aggravate insulin resistance and progression of Diabetic Nephropathy: Case-control study, the authors describe how the expression of MMP2, PTEN, NF-ĸB and MicroRNA-181b-5p in patients with Type 2 diabetic Mellitus correlates with the progression to nephropathy by aggravating the resistance to Insulin, inflammation and renal fibrosis.
Comments to the Authors:
Paragraph 3.1: the parameters are described in the Table S5 and not S4.
Paragraph 3.2: the paragraph is not clear. Please rephrase it.
Paragraph 3.3: for the gene expression is better to use Fold Change instead of Fold Change expression.
Different values are reported in the table S6 compared to the text.
MMP2 values are not significant.
Paragraph 3.4: is better to refer to the figure panel in the text to facilitate the reading and a better description of the panel D (Figure 3) is required.
Paragraph 3.5: the panel 4E needs to be described in the text.
Overall, a revision of the English and formatting is required for the text.
Author Response
Reviewer 4
In the manuscript entitled PTEN, MMP2 and NF-κB and regulating MicroRNA-181 aggravate insulin resistance and progression of Diabetic Nephropathy: Case-control study, the authors describe how the expression of MMP2, PTEN, NF-ĸB, and MicroRNA-181b-5p in patients with Type 2 diabetic Mellitus correlates with the progression to nephropathy by aggravating the resistance to Insulin, inflammation and renal fibrosis.
Response: Thank you for reviewing our manuscript. We appreciated your feedback.
Comments to the Authors:
Paragraph 3.1: the parameters are described in the Table S5 and not S4.
Response: We appreciate the reviewer for bringing to our attention that in Paragraph 3.1, the parameters were actually described in Table S5 and not S4. We apologize for this mistake and have made the necessary corrections to the manuscript. Thank you for helping us improve the clarity and accuracy of our work.
Paragraph 3.2: the paragraph is not clear. Please rephrase it.
Response: Thank you for your feedback on our manuscript. We appreciated your comments and agreed that the clarity of paragraph 3.2 could be improved. In response to your suggestion, We have rephrased the paragraph as follows:
“Compared to HC, T2DM, and DN were found to have significantly higher levels of FBS, HbA1c, insulin, and Insulin and homeostasis model assessment-estimated insulin resistance (HOMA-IR), indicating insulin resistance. In addition, renal function tests such as urea and creatinine were significantly elevated, while GFR (mL/min/1.73 m²) was significantly lower in the DN cases, suggesting renal damage. T2DM and DN patients also had significantly higher levels of LDL (Low-density lipoprotein), HDL (High-density lipoprotein), and triglycerides in their lipid profile compared to HC.”
Paragraph 3.3: for the gene expression is better to use Fold Change instead of Fold Change expression.
Response: Thank you for your feedback on our manuscript. We appreciate your suggestion regarding the use of "Fold Change" instead of "Fold Change Expression" in paragraph 3.3 for gene expression analysis. We agree that "Fold Change" is a more appropriate term and will make the necessary revisions to the manuscript to reflect this change.
Thank you again for your valuable feedback and for helping us to improve the clarity and accuracy of our manuscript.
Different values are reported in the table S6 compared to the text.
Response: Thank you for bringing this to our attention. We apologize for any confusion that may have been caused by the discrepancy in values between the text and Table S6. After a thorough review, we identified an error in the transcription of the data in the table. We have corrected this error in the revised manuscript to ensure that the values reported in the text and tables are consistent and accurate. We appreciate your attention to detail and your help in improving the accuracy of our manuscript. Thank you again for your feedback.
MMP2 values are not significant.
Response: Thank you for bringing this to our attention, and we have corrected the mistake in the revised manuscript that accurately reflects these findings.
Paragraph 3.4: is better to refer to the figure panel in the text to facilitate the reading and a better description of the panel D (Figure 3) is required.
Response: Thank you for your feedback. We have revised the manuscript to refer to the figure panels in the text and have added a more detailed description of panel D in Figure 3.
“Figure 3D presents a radar plot that shows the significant KEGG pathways influenced by the genes that were studied. The values on the plot represent the -log10 FDR (false discovery rate) of the pathways. Two important pathways, endocrine resistance (MMP2) and insulin resistance (PTEN, NF-κB), were found to be influenced by the studied genes. Several other pathways were also found to be influenced by the studied genes, including the T cell receptor signaling pathway (NFKBIA, NFKB1, CHUK, RELA, PI3R1), the PI3K-Akt signaling pathway (NFKB1, TP53, CHUK, PTEN, RELA, PIK3R1), Th1 and Th2 cell differentiation (NFKBIA, NFKB1, CHUK, RELA), the IL signaling pathway (NFKBIA, NFKB1, CHUK, RELA), the AGE-RAGE signaling pathway in diabetic complications (MMP2, NFKB1, RELA, PIK3R1), Endocrine resistance (MMP2, TP53, PIK3R1), and the mTOR signaling pathway (CHUK, PTEN, PIK3R1). Overall, these findings suggest that the studied genes play an important role in various cellular pathways and processes, including insulin resistance, immune cell signaling, cellular growth and differentiation, and renal fibrosis.”
Paragraph 3.5: the panel 4E needs to be described in the text.
Response: Thank you for your feedback. We have revised the manuscript to refer to the figure panels in the text and have added a more detailed description of panel E in Figure 4.
“The study population's GFR was calculated using the MDRD formula and CKD was classified into five stages based on severity. The FC of MMP2 demonstrated a continuous trend of downregulation with the severity of CKD, while PTEN's FC displayed a significant downregulation trend (p=0.05). On the other hand, NF-ĸB expression exhibited a significant upregulation trend (p=0.05) with CKD severity. The FC of hsa-miR-181b-5p showed a significant upregulation trend (p=0.05) with renal damage. A trend plot presented in Figure 4E showed the complete visualization of the FC distribution and the trend of all genes and microRNA in a single graph. The trend analysis revealed that PTEN and MMP2 were downregulated, while NF-ĸB and hsa-miR-181b-5p were upregulated with the progression of CKD. The stages of CKD were categorized based on GFR and the percentage of kidney function, and the mean values for FC of the genes and microRNA were calculated for each stage. The FC of MMP2 remained relatively consistent across all stages, whereas PTEN and NF-ĸB's FC decreased with the progression of the disease.”
Overall, a revision of the English and formatting is required for the text.
Response: Thank you for your review of our manuscript. We appreciate your feedback and agree that the English and formatting of the text could be improved. In response to your suggestions, we carefully revised the manuscript to address these issues. We thank you for your constructive comments and for helping us to improve the quality and readability of our manuscript.
Round 2
Reviewer 1 Report
The authors have successfully addressed all comments